# Gradually Increasing the Temperature Reduces the Diapause Termination Time of *Trichogramma dendrolimi* While Increasing Parasitoid Performance

**DOI:** 10.3390/insects13080720

**Published:** 2022-08-11

**Authors:** Xue Zhang, Bingxin He, Lucie S. Monticelli, Wenmei Du, Changchun Ruan, Nicolas Desneux, Junjie Zhang

**Affiliations:** 1Engineering Research Center of Natural Enemies, Institute of Biological Control, Jilin Agricultural University, Changchun 130118, China; 2Université Côte d’Azur, INRAE, CNRS, UMR ISA, 06600 Nice, France

**Keywords:** *Trichogramma*, diapause development, fluctuating temperature, biological control

## Abstract

**Simple Summary:**

*Trichogramma dendrolimi* is one of the most widely used biological control agents around the world, and diapause is a vital way to preserve *Trichogramma* products during mass production. However, diapause is a time-consuming process. In this study, we evaluated the effect of gradually increasing temperature on the diapause termination and the quality of *Trichogramma* products. The diapause termination rate reached the 95% required by practical applications when the induction period was high or in the treatments involving at least two-step variable temperatures (preceded by an induction period of 55, 60, and 70 days). Moreover, treatments consisting of at least two different temperatures led to higher parasitism and emergence rates while keeping the other parameters constant. In addition, the treatment with the highest temperature variation preceded by only 55 days of induction period had the highest population trend index. Our results demonstrate that gradually increasing temperature allows *T. dendrolimi* to complete diapause earlier than at present while increasing its potential pest control capacity and providing additional flexibility in mass production of *T. dendrolimi*.

**Abstract:**

*Trichogramma dendrolimi* Matsumura is widely used as a biological control agent of many lepidopteran pests. Diapause has been used as an effective method to preserve the *Trichogramma* products during mass rearing production. However, it currently takes at least 70 days to break diapause, and we tested whether gradually increasing the temperature instead of using constant temperature could reduce the time of diapause termination and offer a higher flexibility to *Trichogramma* producers. The diapause termination rates of individuals kept at different conditions were measured, and five groups for which diapause termination rate reached the 95% were selected to test five biological parameters, including the number of eggs parasitized, the parasitism and emergence rates, the female sex ratio, the wing deformation rate, and the parasitoid longevity. Compared to the currently used procedure (70 days at 3 °C), treatments with at least two different temperatures resulted in higher parasitism and emergence rates while keeping the other parameters constant. The treatment that consisted of at least two different temperatures preceded by only 55 days of induction period had the highest population trend index, meaning that the population under these conditions grows more rapidly. Our results demonstrate that gradually increasing temperature allows *T. dendrolimi* to complete diapause earlier than at present while increasing its potential pest control capacity and providing additional flexibility in mass production of *T. dendrolimi*.

## 1. Introduction

Diapause is a vital adaptive strategy to cope with adverse environmental conditions in insects [1]. In insects, it is a temporary interruption of development, which possesses two notable characteristics: low metabolic activity and extended longevity [2,3,4,5,6] in mass rearing production of biocontrol agents such as *Trichogramma* sp., which is one of the most widely used egg parasitoids to control multiple major agricultural pests [7,8,9,10]. *Trichogramma dendrolimi* is one of the most successful commercialized natural enemies of insects [7,11,12] and was applied via inundative release across 200,000–350,000 ha of corn crops annually in the 1980s and 1990s, which later increased to 2.3 million ha annually since 2012 in Jilin province to control corn pests such as Asian corn borer (ACB), *Ostrinia furnacalis* (Guenee) [9]. Diapause has been used as an alternative to cold storage. Diapause can extend parasitoid shelf life from 15 days in cold storage to more than three months. Moreover, diapause increases emergence rate [13,14] and female fecundity [15], which in turn results in better management of biocontrol agent production and their efficiency.

Insect diapause is a dynamic process consisting of several continuous phases: pre-diapause (diapause induction and diapause preparation), diapause (diapause initiation, diapause maintenance, and diapause termination), and post-diapause development (resume development after diapause termination). Diapause is induced only when particular environmental thresholds are reached [16,17,18]. In *Trichogramma*, almost all the species enter a facultative diapause as prepupa depending on the prevailing environmental conditions [19]. The induction of *Trichogramma* diapause has been intensively studied to determine the optimized conditions that contribute to preserve *Trichogramma* products in a good quality for a long period [13,19,20]. Based on these studies, temperature has been shown to play a strong role in diapause induction [13,21,22] and varies according to the species considered. For example, the optimum temperature for both *T. cordubensis* and *T. carverae* to induce diapause was 10 °C [13,23]. In *T. dendrolimi*, the proportion of diapause was highest at 12 °C [24]. In contrast, even though diapause termination and post-diapause development are also important steps in diapause development, they have been scarcely studied.

Low temperatures in winter play an important role in diapause termination under natural conditions [25,26]. In laboratory conditions, appropriate chilling temperature and duration is important for successful diapause termination [19,27,28]. For example, *Telegrayllus emma* could terminate diapause after chilling at 8 °C for more than 40 days [29]. In practical application, diapause development and especially diapause termination are time-consuming processes. For *Trichogramma* species, it may take at least 70 days to break diapause, with the addition of diapause induction, which represents an approximately 100-day shelf life that is much longer than the required time for mass production [19]. To reduce the termination time, a gradual increase of the temperatures, as found in natural conditions, may be used to simulate the recovery of optimal development conditions for parasitoids. Indeed, large-scale industrialized production of *Trichogramma* spp. is currently using a single and constant temperature (1 or 3 °C) to terminate the diapause [19,28] while the temperatures in the wild are gradually increasing and even vary within a day (higher temperature during the day and lower during the night). This suggests that the application of intermediate temperature steps can shorten the time of diapause termination while maintaining individual performance (i.e., parasitism and emergence rates as emergence rate should be higher than 95% to ensure the economic profitability of the mass production).

To test the hypothesis that gradually increasing the temperature could reduce the time of diapause termination, we tested 25 different conditions varying in terms of (i) induction period (from 40 to 70 days), (ii) number and duration of temperature steps (1 or 2 intermediate temperatures during 2 or 5 days), and (iii) temperatures within a day (20:15 and/or 25:20 °C during 16:8 h L:D) on the ability of the parasitoid *T. dendrolimi* to terminate diapause. We then selected five promising conditions to test their impact on the parasitoid performance represented in this study by the number of eggs parasitized, the parasitism and emergence rates, the female sex ratio, the deformation rate, and the parasitoid longevity. In addition, we calculated an index of population trends to estimate population dynamic when parasitoids encounter these conditions. It may help to evaluate the management effectiveness of biological agents at generation level and provides an integrated index containing all regulation effects information of environmental factors on the whole generation of the target population [30,31].

## 2. Materials and Methods

### 2.1. Biological Material

The original population of *T. dendrolimi* was collected with eggs of *Ostrinia furnacalis* from corn fields in Binxian, Heilongjiang province, China (45°45′ N, 127°30′ E) in 2018. The specimens were identified using SEM micrographs by male genital [32]. The colony of *T. dendrolimi* was reared exclusively on eggs of the tussah silkworm, *Antheraea peryni*, for more than five generations. The conditions for parasitoid rearing were 26 ± 1 °C, 70 ± 5% RH, and a 16:8 L: D photoperiod.

Egg masses of *A. pernyi* were purchased in Yongji, Jilin province, China (126°31′ E, 43°38′ N), and fresh eggs of *A. perny*i were obtained via dissecting female ovaries [33]. After cleaning with distilled water, the low-quality eggs were removed and the qualified eggs were dried at ambient temperature.

### 2.2. Experiment 1: Diapause Termination Rate

Approximately 2000 *A. pernyi* eggs were parasitized for 8 h by about 5000 adults of *T. dendrolimi* under condition of 26 ± 1 °C, 70 ± 5% RH, with a 0:24 h L:D photoperiod. Since light affects the efficiency of inducing diapause, we performed it in the darkness. The parasitized host eggs were then kept for approximately 53 h at laboratory conditions (26 ± 1 °C, 70 ± 5% RH, and a 16:8 h L:D photoperiod) until the parasitoids developed to the middle larval stage (after rearing about 53 h, ten parasitized *A. peryni* eggs were dissected and the parasitoids within the eggs were observed under binocular). The specific method of diapause induction for *T. dendrolimi* reared with *A. pernyi* eggs was followed by Zhang et al. [24]. The diapaused parasitoids remained at the prepupal stage, not continuing to develop.

After being kept for 30 days at 12 °C to induce diapause [24], the host eggs were randomly assigned to five different groups and treated at 3 °C for 40, 50, 55, 60, or 70 days (currently, the optimal conditions used in mass rearing production for diapause termination of *T. dendrolimi* is 3 °C for 70 days). After these periods, parasitoids from each group were randomly assigned to five different treatments: 1 = 25 °C until emergence; 2 = 10 °C for 2 days followed by 25 °C until emergence; 3 = 10 °C for 5 days followed by 25 °C until emergence; 4 = 10 °C for 2 days followed by 20:15 °C for 16:8 h per day during to 2 days then to 25 °C until emergence; 5 = 10 °C for 2 days followed by 20:15 °C for 16:8 h per day for 2 days, then 25:20 °C for 16:8 h per day until emergence (Table 1). After these second periods, we counted the number of adults until no more adults emerged and dissected the eggs using binoculars to count the number of pupae and the number of prepupae. Fifteen eggs per conditions (group x treatment) were assigned per replication, and 3 replications were performed (for a total of 45 repetitions (eggs) per condition).

The diapause termination rate has been calculated as follows:$$\;\mathrm{Diapause}\;\mathrm{termination}\;\mathrm{rate}\;\left(\%\right)=\frac{\mathrm{number}\;\mathrm{of}\;\mathrm{adults}\;+\;\mathrm{number}\;\mathrm{of}\;\mathrm{pupae}}{\mathrm{number}\;\mathrm{of}\;\mathrm{adults}\;+\;\mathrm{number}\;\mathrm{of}\;\mathrm{pupae}\;+\;\mathrm{number}\;\mathrm{of}\;\mathrm{prepupae}}\times 100$$

### 2.3. Experiment 2: Performance of Emerged T. dendrolimi

Five conditions (group x treatment), including each treatment (1 to 5) with the highest diapause termination rate identified from 3 groups (55, 60, and 70 days at 3 °C) with the highest rates in the previous experiment (see results) were selected to test the performance of *T. dendrolimi* individuals. *Corcyra cephalonica* (Lep: Pyralidae) has been used as a host in this experiment to measure number of eggs parasitized. *C. cephalonica* was reared in an air-conditioned insectary on artificial diet at (26 ± 1) °C, (80 ± 5)% RH, and 14:10 h L:D photoperiod. A total of 200 fresh eggs of *C. cephalonica*, treated with ultraviolet irradiation for 30 min, were glued on an egg card strip with non-toxic glue (0.9 cm × 0.5 cm). One egg card was then introduced in a glass tube and one single female wasp belonging to one of the five conditions tested was allowed to parasitize the egg for eight hours at (26 ± 1) °C, (80 ± 5)% RH, and 14:10 h L:D photoperiod. After this period, eggs were kept in the same conditions until parasitoid emergence to count the number of host eggs that have become black (a sign of parasitism), the number of host eggs with a hole, the number of females and males emerged, the number of individuals with wing deformation, and the number of unemerged parasitoids after dissection using binoculars. To assess the parasitoid longevity after diapause termination, individuals were kept at (26 ± 1) °C, (80 ± 5)% RH, and 14:10 h L:D photoperiod and were checked daily to count the number of days they survived. Individuals that did not enter diapause (non-induced) were used as control. Fifteen bioassays were performed per condition (group × treatment + control) and 3 replications were performed (for a total of 45 repetitions per condition).

The biological parameters evaluated have been calculated as follows:$$\;\mathrm{Parasitism}\;\mathrm{rate}\;\left(\%\right)=\frac{\mathrm{number}\;\mathrm{of}\;\mathrm{host}\;\mathrm{eggs}\;\mathrm{with}\;a\;\mathrm{hole}}{\mathrm{total}\;\mathrm{number}\;\mathrm{of}\;\mathrm{host}\;\mathrm{eggs}}\times 100$$
$$\;\mathrm{Female}\;\mathrm{sex}\;\mathrm{ratio}\;\left(\%\right)=\frac{\mathrm{number}\;\mathrm{of}\;\mathrm{female}\;\mathrm{adults}}{\mathrm{total}\;\mathrm{number}\;\mathrm{of}\;\mathrm{offsprings}\;\mathrm{emerged}}\times 100$$
$$\;\mathrm{Deformation}\;\mathrm{rate}\;\left(\%\right)=\frac{\mathrm{number}\;\mathrm{of}\;\mathrm{deformed}\;\mathrm{adults}}{\mathrm{total}\;\mathrm{number}\;\mathrm{of}\;\mathrm{offsprings}\;\mathrm{emerged}}\times 100$$
$$\;\mathrm{Emergence}\;\mathrm{rate}\;\left(\%\right)=\frac{\mathrm{number}\;\mathrm{of}\;\mathrm{emerged}\;\mathrm{adults}}{\mathrm{total}\;\mathrm{number}\;\mathrm{of}\;\mathrm{offsprings}}\times 100$$

The index of population trend (*I*) described by Morris (1963) was calculated as: *I* = *S_P_* × *FP_F_* × *P**_♀_*, where *S_P_ is* the parasitism rate as describe above (standing for the survival rate of the egg to the adulthood), *FP_F_* is the number of eggs parasitized per female, and *P**_♀_* is the female proportion [34].

## 3. Statistical Analysis

We used generalized linear mixed models (GLMMs, package ‘LmerTest’) following a binomial distribution to analyze the diapause termination rate, the parasitism rate, the female sex ratio, the wing deformation rate, and the emergence rate. Linear Mixed Models (LMM) have been used to analyze the number of host eggs parasitized and the population trend index (I) in the experiment two as the model residuals followed a normal distribution (visual quantile–quantile plot). In experiment one, the fixed factors were the group and the treatment. In experiment two, the fixed factor was the condition (group x treatment) as the group and the treatment selected were unbalanced. The replication has been implemented as a random effect in all the models. Statistical significance of the group, the treatment, their interaction, or the condition was determined by analysis of variance (ANOVA) with an χ^2^ test (package ‘car’). When a factor or an interaction had a significant effect, multi-comparison tests were performed using the ‘multcomp’ package to compare each factor or condition with each other, respectively. The impact of the condition on the parasitoid longevity was analysed using a Cox model (Kaplan–Meyer Log rank, package ‘Survival’). All the statistical analyses were performed in RStudio on 2 September 2021.

## 4. Results

### 4.1. Experiment 1: Diapause Termination Rate

The diapause termination rate varied significantly depending on the group (X^2^_4_: 21,116.9, *p* < 0.001), the treatment (X^2^_4_: 10,464.8, *p* < 0.001), and their interaction (X^2^_16_: 1174.9, *p* < 0.001) (Figure 1). The rate of diapause termination was 3.3 times higher when individuals stayed 70 days at 3 °C than when they stayed only 40 days. After 40 days at 3 °C, treatments 4 and 5 were enabled to reach the highest diapause termination rate, but it was up to 45% on average. In the group where individuals stayed 50 and 70 days at 3 °C, treatments 3 and 5 were enabled to reach, on average, 80% and 98% of the diapause termination rate, respectively. After 55 and 60 days, the highest diapause termination rate reached an average of 98% in treatments 4 and 5. In treatments 1 and 2, the proportion of diapause termination was over 95% only when the parasitized eggs were kept at 3 °C for 70 days. In treatments 3 and 4, this proportion was over 95% when the parasitized eggs were kept at 3 °C for 60 and 70 days. In treatment 5, this proportion was overcome after 55, 60, and 70 days at 3 °C. According to these results, the five conditions including each treatment from a different group reaching 95% of diapause termination rate have been selected to evaluated the *T. dendrolimi* performance: (i) individuals were kept for 70 days at 3 °C, then following treatment 1, (ii) individuals were kept for 70 days at 3 °C, then following treatment 2, (iii) individuals were kept 60 days at 3 °C, then following treatment 3, (iv) individuals were kept for 60 days at 3 °C, then following treatment 4, and (v) individuals were kept for 55 days at 3 °C then following treatment 5. The significant impact of the interaction indicates a strong effect of the treatment on the rate of diapause termination when the induction period was between 40 et 55 days, whereas this effect was low when the induction period was 70 days.

### 4.2. Experiment 2: Performance of Emerged Trichgoramma dendrolimi

The performance of the parasitoids was studied through the evaluation of the number of parasitized eggs, the rate of parasitism (proportion of eggs with a hole), the rate of emergence (proportion of individuals having emerged), the sex ratio, the rate of wing deformation of emerged adults, and the adult longevity.

The number of eggs parasitized varied depending on the condition (X^2^_5_: 19.9, *p* = 0.001) and was 1.2 times higher when the condition included the treatments 2, 3, and 5 compared to the control (Figure 2A). The parasitism rate also varied depending on the condition (X^2^_5_: 322.4, *p* < 0.001) and was higher in the control and the treatment 4 and 5 compared to the treatments 1 and 2 (Figure 2B). The emergence rate was above 80% and slightly higher in the control and the treatments 2, 4, and 5 than in the treatments 1 and 3 (X^2^_5_: 46.8, *p* < 0.001, Figure 2C). By contrast, the deformation rate was lower in the control compared to all the conditions tested and was the highest (reaching 15%) in the conditions including the treatment 1, 3, and 5 (X^2^_5_: 380.0, *p* < 0.001, Figure 2E). The female sex ratio (X^2^_5_: 9.3, *p* = 0.099) and the longevity (Logrank test: 1.2, *p* = 0.900) did not vary according to the conditions tested (Figure 2D,F).

The index of population trend (*I*) varied significantly among the conditions (X^2^_5_: 223.2, *p* < 0.001, Table 2). The highest *I* value was found in the condition including treatment 5 and was 1.15 times higher than the conditions including treatments 2, 3, and 4, and 1.3 than the condition including treatment 2 and the control (Table 2).

## 5. Discussion

Storage time plays an important role in industrial production of *Trichogramma* parasitoids as longer storage time enables flexible supply of parasitoids for timely and urgent field release in biological control programs. The most commonly used preservation method for *Trichogramma* is cold storage [35]. However, this method not only cannot guarantee the shelf life, but also affects the quality of natural enemy products. For example, the survival rate of *T. dendrolimi* sharply decreased after being saved under low temperature for more than 3 weeks [21]. In our previous studies, we illustrated that diapause was a better method for *T. dendrolimi* preservation, which could greatly extend shelf life from approximately 15 days to over 3 months [15,19]. In this study, and, in comparison to the current recommended method to induce diapause termination (kept individuals at 3 °C for 70 days), the individuals kept at 3 °C for 55 and 60 days and then kept at various temperatures gradually increasing for 4 days reached the 95% of diapause termination rate required by the *Trichogramma* producers. In addition, those individuals exhibited higher parasitism and emergence rates while keeping the other parameters constant. The calculation of the population trend index indicated that the treatment keeping individuals at 3 °C for 55 days and then at 10 °C for 2 days followed by 20:15 °C for 16:8 h per day during to 2 days then 25:20 °C for 16:8 h per day until emergence was the more optimal treatment. This demonstrated that gradually increasing temperature allows *T. dendrolimi* to complete diapause 11 days earlier while increasing its performance as a biological control agent.

The interaction between the induction time of diapause termination and the treatment (whether gradually increasing the temperature) was very strong with a positive impact on the higher diapause termination rate of treatments when the induction period was longer. In many insects, low-temperature stimulation was essential for diapause termination; without low temperature, insects remain in the diapause stage until they die [19,25]. In order to complete diapause development, a long enough chilling period must be provided [36]. For *Rhagoletis* species, there is no adult emergence from puparia with an insufficient chilling period [37,38]. Similarly, for *Colpoclypeus florus*, chilling temperature and duration of chilling significantly affected diapause termination. When the chilling period was long enough, the diapause development proceeds rapidly [39].In fact, termination of diapause is a dynamic developmental process, which is subdivided into two parts, endogenous and exogenous diapause [16]. Endogenous diapause is defined as the part of diapause when insects are non-responsive to external conditions in the maintenance of the diapause phenotype, while during exogenous diapause, insects can respond developmentally to external conditions. Once the insect transits to exogenous diapause, increasing temperatures lead to post-diapause development and the emergence of a new generation of reproductively competent adults. Additionally, chilling was needed for diapause to progress from endogenous diapause to post-diapause quiescence [40].

In *T. dendrolimi*, when the induction time at 3 °C was only 40 days, the diapause termination rate was low regardless of subsequent treatment. However, when the induction time was 70 days, the diapause termination rate under all treatments was high. As chilling time increased, diapause intensity decreased. It showed that 40 days of chilling was not enough for transition *T. dendrolimi* from endogenous to exogenous diapause. This was similar to the result obtained in *R. cerasi*, namely insects could have a different response to insufficient chilling and extended exposure to chilling [38]. Interestingly, the treatments had a much bigger effect on the diapause termination rate when the induction time was short. Among different treatments, constant or fluctuating temperatures were used for post-diapause development. According to a previous study, the developmental rate of insects was strongly temperature-dependent. Furthermore, fluctuating temperature was more influential on developmental rate than that under constant temperature [41].

In the post-diapause stage, and, in comparison to the condition of constant temperature (70 days at 3 °C), the gradual increase of temperature enabled an increase in the rate of parasitism and emergence but not the number of eggs parasitized, the female sex ratio, the wing deformation rate, and the longevity. Metabolic depression is a highly conserved feature of inset diapause [42]. Ragland et al. (2009) reported the metabolic rate of diapausing *R. pomonella* puparia to be only 7.5% of that for puparia in the post-diapause stage [43]. Additionally, an increase in metabolism is a reliable indicator of diapause termination and the initiation of post-diapause development. However, it needs time to resume metabolism during the completion of diapause development. Metabolic rate changes dynamically as diapause progresses, reaching a minimum several weeks after onset, followed by a gradual increase towards the time of diapause termination. In *R. pomonella*, after a transition from 4 to 24 °C, the metabolic rate of puparia did not increase for 10 days [43]. Most studies dealing with the effects of temperature on insects have been conducted at constant temperatures, which are not comparable with field conditions. Natural conditions are characterized by daily thermal cycles, and temperature changes at night might affect the biology of insects. Therefore, fluctuating temperatures may provide a more accurate assessment of natural enemy insects than constant temperatures. Our study revealed significant differences between fluctuating and constant temperature effects on several biological parameters of *T. dendrolimi*. The parasitism rate and emergence rate under fluctuating temperature were higher than those under constant temperature. These results were comparable to those of Delava et al. (2016), who found that *Leptopilina boulardi* had a greater parasitism rate under fluctuating temperatures [44]. Moreover, the fluctuating temperature was more suitable than the constant temperature for rearing *Telenomus podisi* [45].

However, some of the measured parameters were not affected by fluctuating temperatures, possibly because of the fluctuating temperature regimes used in this study (20:15 °C for 16:8 h per day, the average temperature was 18.3 °C; 25:20 °C for 16:8 h per day, the average temperature was 23.3 °C) did not provide average temperature levels exceeding threshold temperature for population development and parasitism activity [46]. Exposure to prolonged constant low temperatures has been studied in many other insects and shows a detrimental effect on the survival of parasitic wasps [47,48,49,50]. Low temperatures can induce lesions, and several types of metabolic waste products such as lactic acid and nitrogenous substances can accumulate to toxic concentrations [51]. In the present study, the parasitism rate and emergence rate were improved when the *Trichogramma* individuals receive two fluctuating temperature buffering stages. It can be inferred that transferring wasps to fluctuating temperatures (from low to high) may reduce the speed and the amounts of accumulated injuries; namely, the substantial part of the accumulated toxins was eliminated. As Colinet et al. reported, accumulated injuries may be repaired every day under the fluctuating temperature [52]. Positive parental effects on predictable fluctuating temperatures have also been found in parasitoids, such as an increase in body size and lipid reserves of the offspring when the parental generation was exposed to low temperatures [53]. In addition, fluctuating temperature also regulated the activity of the enzymes. For example, in the Khapra beetle, *Trogoderma granarium*, the individuals under fluctuating temperatures had the highest changes in the activity of the enzymes, which was significantly higher than that under rapid cold-hardened conditions [54].

Inducing diapause is generally believed to be costly. Such costs are usually described in the form of existence of trade-offs between diapause duration and other life-history traits. Individuals experiencing diapause have lower fecundity and shorter adult longevity compared with non-diapause individuals [55,56]. However, in a few insect species, diapause has a positive effect on post-diapause adult life-history traits, especially the fecundity. For example, in the grasshopper *Tetrix undulata*, *Grapholitha funebrana*, and cabbage beetle *Colaphellus bowringi*, females that experience diapause had greater fecundity [57,58,59]. In the corn stalk borer *Sesamia noagrioides*, the fecundity was also positively correlated with the duration of diapause [60]. By contrast, diapause has no or little effect on total fecundity of the Colorado potato beetle, *Leptinotarsa decemlimeata* [61,62]. Therefore, the correlation between diapause and fecundity varies with insect species. Additionally, in *T. dendrolimi*, fecundity was improved after experiencing diapause development [15], whereas the mechanism still remains unclear. Similar results were gained in other *Trichogramma* species used for controlling sugarcane borer. In a field test, the parasitism of sugarcane borer eggs in the field where diapaused *T. ostriniae* was released was significantly higher than that of non-diapause wasps [63]. During diapause development, the content of trehalose increased significantly. Trehalose, the major insect blood sugar, plays an important role in the diapause of many insects, serving as an energy source and a stress protectant [64]. In *T. dendrolimi* and *T. chilonis*, the fecundity was remarkably improved after feeding trehalose (unpublished data). Therefore, in *T. dendrolimi*, trehalose accumulated during diapause may not only serve as cryoprotectants, but also as energy reserves for post-diapause development and oviposition. Wang et al. (2011) tested the contents of glycerin, protein, and total carbohydrate during diapause and post-diapause development [65]. The results indicated that the content of total carbohydrate of individuals experiencing diapause development was always higher than that in non-diapause individuals [65]. So, besides trehalose, other kinds of carbohydrate may also play a certain role in regulating the fecundity of *Trichogramma* parasitoids that experienced diapause development.

The index of population trend (I) provides an integrated index containing all regulation effects information of environmental factors on the whole generation of the target population, and this index can be the analytical tool to evaluate the management effectiveness of biological agents at generation level [30]. The index of population trend was a mathematical model of population trend, and it used to predict the insect’s population growth under ideal conditions. In this formula, the deformed females were included in the total female. The deformed female was the individual for which wings were not as spread as the normal one, whereas the parasitic ability was affected little. Certainly, if the deformed females were released into the field, the mobility could be a problem that needs to be discussed further.

Further extension of shelf life is another form of flexibility in the mass production of *Trichogramma* products. Though some kinds of insects could prolong longevity under low temperatures even for several years, for *Trichogramma*, three months is a limit to keep high control efficiency. In using *Trichogramma* to control pests, the most important thing is to release the wasps at the same time as pests lay eggs. For many pests, the duration of the egg stage is quite short. It is vitally important to release natural enemies at the optimal time. However, because of the instability and unpredictability in the field, sometimes we need to release wasps ahead of schedule. We need a method to prolong the shelf life, while we also need a method to break it quickly, thereby slowing development (inducing diapause) and giving producers flexibility (break diapause) in timing emergence to weather conditions or pest spawning peak. Nowadays, *Trichogramma* parasitoids have been used mostly through inundative releases [9]. In the application of *Trichogramma*, it is impossible to produce a huge number of parasitoids for a large area where there is an urgent need to control pests within a short time. Therefore, the flexibility of utilizing diapause technology was essential.

In conclusion, effective chilling is required to break diapause in *T. dendrolimi*, thus enabling parasitoids to reach the next developmental stage. In addition, fluctuating temperature appears more suitable for the post-diapause development of *T. dendrolimi*. Successful biological control requires accurate knowledge on mass rearing and optimal usage conditions of the control agents to ensure effectiveness [8,9,10,66,67]. From our study, we provided an alternative choice in the process of industrialized production. To maximize the shelf life, producers can increase chilling duration, but if the parasitoids needed to be released in advance, the method in the present study can greatly reduce the time to terminate diapause. Furthermore, the quality of the natural enemy products can be guaranteed or even improved, enabling building more effective, sustainable, and environmentally sound Integrated Pest Management packages [68,69], and thus lowering multiple potential negative effects of chemical pesticides on beneficial organisms [70].

## Figures and Tables

**Figure 1 insects-13-00720-f001:**
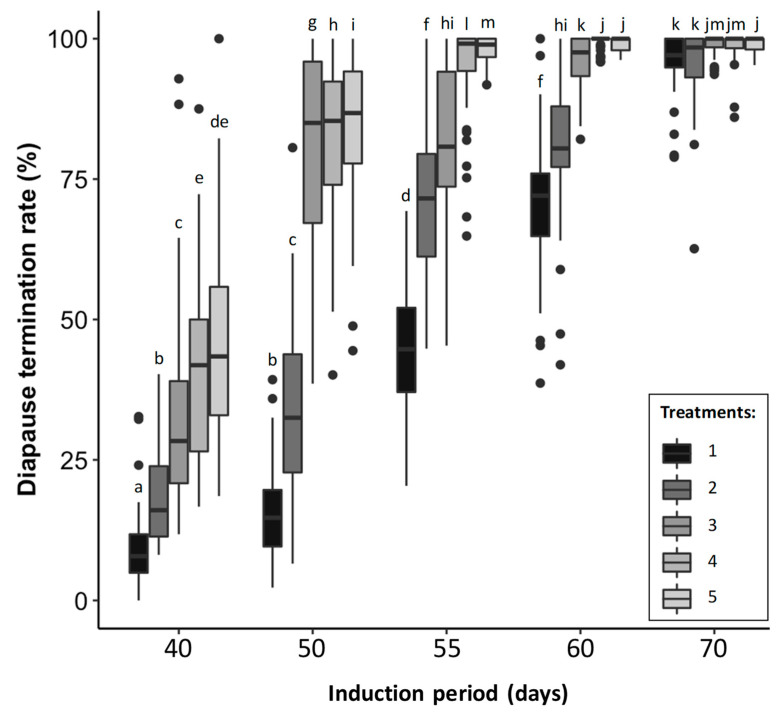
Diapause termination rate (%) in *T. dendrolimi* depending on the group (diapause induction at 3 °C for 40, 50, 55, 60, or 70 days) and the treatments described in Table 1. Different lower-case letters on top of the boxplots indicated a significant difference among conditions (each treatment belonging to each group) (*p* < 0.05). The black circles represent the outliners and what the letters represent is described in the legend. Different letters indicate a statistical difference for each condition with the others (GLMM following a binomial distribution).

**Figure 2 insects-13-00720-f002:**
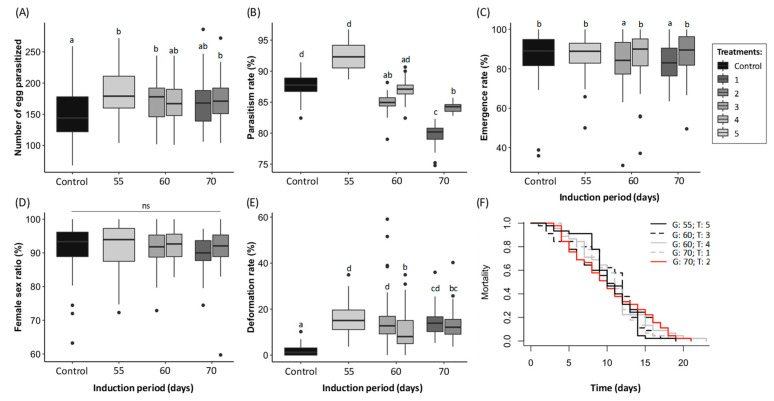
(**A**) Number of egg parasitized (turn black), (**B**) parasitism rate (proportion of egg with a hole), (**C**) emergence rate (proportion of adult emerged), (**D**) female sex ratio, (**E**) deformation rate of adult*,* and (**F**) longevity (days) of *T. dendrolimi* depending on the group (diapause induction at 3 °C for 55, 60 or 70 days) and the treatment (1): 25 °C until emergence; 2:10 °C for 2 days followed by 25 °C until emergence; 3:10 °C for 5 days followed by 25 °C until emergence; 4:10 °C for 2 days followed by 20:15 °C for 16:8 h per day for 2 days, then 25 °C until emergence; 5:10 °C for 2 days followed by 20:15 °C for 16:8 h per day for 2 days, then 25:20 °C for 16:8 h per day until emergence). The control corresponds to individuals that did not enter diapause (not induced). Different letters on top of the boxplots indicated a significant difference among conditions (each treatment belonging to each group) for each biological parameter (*p* < 0.05). ns: non-significant. The black circles represent the outliners and what the letters represent is described in the legend. Different letters indicate a statistical difference for each condition with the others (GLMM following a binomial distribution for rates, LMM for the number of parasitized eggs and a Cox model for mortality).

**Table 1 insects-13-00720-t001:** The different treatments for diapause termination of *T. dendrolimi* reared on *A. pernyi* eggs.

Induction	Treatment	Buffering	Development
3 °C for 40, 50, 55, 60, 70 (d), respectively, then divided into five treatments	1	--	25 °C until to adults
	2	10 °C (2d)	25 °C until to adults
	3	10 °C (5d)	25 °C until to adults
	4	10 °C (2d)-20-15 °C (2d)	25 °C until to adults
	5	10 °C (2d)-20-15 °C (2d)	25-20 °C until to adults

Note: The photoperiod when the host eggs kept at 3 °C was 0:24 h L:D, and the other is 16:8 h L: D. 20-15 °C means the host eggs are kept at 20 °C for 16 h and 15 °C for 8 h. 25-20 °C means the host eggs are kept at 25 °C for 16 h and 20 °C for 8 h.

**Table 2 insects-13-00720-t002:** The index of population trend of *T. dendrolimi* terminated diapause of five selected treatments.

Index	T1	T2	T3	T4	T5	CK
*S_p_*	0.797	0.841	0.850	0.870	0.924	0.877
*FP_F_*	168.556	171.578	171.067	168.600	181.289	147.822
*P_♀_*	0.826	0.854	0.840	0.843	0.844	0.844
*I*	111.1 a	123.3 b	122.1 b	123.6 b	141.5 c	109.4 a

Note: *S_p_*: parasitism rate of parasitoids (proportion of eggs with a hole), *FP_F_*: the number of eggs parasitized per female, *P**_♀_*: the female proportion. The I index followed by the same letter are not significantly different (*p* > 0.05). The descriptions of five selected treatments (T1–T5) can be seen in caption to Figure 2.

## Data Availability

The datasets generated during the study are available from the corresponding author on reasonable request.

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
