# Peer review of "Gradually Increasing the Temperature Reduces the Diapause Termination Time of Trichogramma dendrolimi While Increasing Parasitoid Performance"

_insects, 2022, doi:10.3390/insects13080720_

Round 1

Reviewer 1 Report

SIMPLE SUMMARY

-Lines 12-14: It should be „….we evaluated effects of …….not only on the diapause termination ….bu also onthe quality…“

-Line 14: Required by whom?

-Line 16: Do not start the sentence with „And“. Please, check this in other parts of the text.

-Line 18: Is it „with the highest temperature variation“? I also think that „occuring“ can be deleted.

ABSTRACT

-Lines 28-31: The sentence is quite confused. Please explain in this sentence what were the experimental groups.

-Line 32: Put „wing deformation rate“.

-Line 33: It should be „used“. Maybe you can use „protocol“ or „procedure“ instead of „parasitoid condition“.

-Lines 35-36: Put „highest“ instead of „most“. Delete „occuring“.

INTRODUCTION

-Line 56: Do you mean „specimens“ or „individuals“ instead of „species“?

-Lines 63-66: The sentence is confused.

-Lines 68-69: Please rewrite this sentence. Apropriate chilling temperature and duration is important for successful diapause termination.

-Line 93: Replace „suffering“ with „encounter“.

MATERIALS AND METHODS

-Line 125: Parenthesis is missing. Delete comma after „days“.

-Line 147: „to measure number of eggs parasitized” is in larger font.

-Line 159: How did you feed adults?

-Lines 160-162: Please be more clear.

-Formulas are in larger font in lines 165 and 166 than in 164 and 167.

-Statistical analysis: Font and line space are not in accordance with template.

-Line 174: „wing deformation“.

RESULTS

-Line 212-216: I suggest you to put „treatments described in Table 1“ after „and“. Then, you can omit detailed description of treatments.

-Lines 239-240: Mention that T1-T5 are 5 selected treatments and that their descriptions can be seen in caption to Figure 2.

DISCUSSION

-Line 287: Put „transit“ instead of „transition“.

-Lines 293-295: The sentence is not clear. I am not sure that „namely“ is correct term. Maybe you can start new sentence with „Namely“.

-Line 295: Put „…for transition of T. dendrolimi…“.

-Line 308: Put „reported“ instead of „report“.

-Line 315: Reference 42 is cited in red color.

-Lines 358-360: Do you mean „….was significantly higher than…“?

-Line 376: „…it is used to predict…population growth …“

-Line 377: „….deformed females were….“

-Lines 379-380: „….females were released……that is needed to duscuss….“

-Line 385: „…at the same time…“

-Line 399: „….to exogenous diapause. …..“

-Lines 404-410: The sentence is nice but I suggest you to split it into two sentences.

REFERENCES

-Some references have article titles with each word capitalized (e.g., references 2,3,9…).

-Line 431: „…..in newly….“

-Line 444: Trichogramma should be in italic.

-Some journal titles are not abbreviated (references 34,53)

-Line 559: Tuta absoluta should be in italic.

Author Response

Dear reviewer:

Thank you for your valuable comments. We have revised our manuscript. 

Author Response

(The authors gave the same response as above.)

Reviewer 3 Report

This article by Xue Zhang and colleagues (insects-1808792) is well motivated, the structure is appropriate, and the manuscript is well written without missing any key details. The methods used are appropriate for the objectives of the work and, in general, well depicted. The resulting figures are sufficient, informative, and of good quality helping to follow the reasoning throughout the manuscript. The discussion of results and comments on future research was nicely done and will be useful to others. Overall, I enjoyed reading the manuscript. A few minor remarks have been made below for authors to consider.

Some of the authors statements on lines 327-344 would be much stronger if they tie their work to the body of literature that has built up on the bioecology of mass-reared endo- and ectoparasite biocontrol agents (BCAs) under constant and fluctuating temperature regimes in California, USA. They all point to the same direction and could be paired back to this study. Some examples are J. Econ. Entomol. 112: 1560-1574 (mass produced ectoparasite BCAs) or J. Econ. Entomol. 112:1062-1072 (mass produced endoparasite BCAs), but there are others too. Adding these details will improve the paper in my opinion. 

Good luck!

Author Response

(The authors gave the same response as above.)
